# Superparamagnetic Iron Oxide Used Alone Is Non-Inferior to the Combination of Radioactive Tracer and Superparamagnetic Iron Oxide in Detecting Sentinel Lymph Nodes in Early-Stage Vulvar Cancer

**DOI:** 10.3390/cancers17233722

**Published:** 2025-11-21

**Authors:** Marcin A. Jedryka, Tymoteusz Poprawski, Krzysztof Grobelak, Piotr Klimczak, Rafał Matkowski

**Affiliations:** 1Department of Oncology, Faculty of Medicine, Wroclaw Medical University, 50-367 Wroclaw, Poland; rafal.matkowski@umw.edu.pl; 2Department of Oncologic Gynecology, Lower Silesian Oncology, Hematology and Pulmonology Center, 53-413 Wroclaw, Poland; tymoteusz.poprawski@dcopih.pl (T.P.); krzysztof.grobelak@dcopih.pl (K.G.); 3Department of Gynecology and Obstetrics, Provincial Specialist Hospital, 64-100 Leszno, Poland; petros69@op.pl; 4Breast Unit, Lower Silesian Oncology, Hematology and Pulmonology Center, 53-413 Wroclaw, Poland

**Keywords:** superparamagnetic iron oxide, technetium-99, sentinel lymph node, tracer, nanoparticle, vulvar cancer

## Abstract

Sentinel lymph node (SLN) detection is a recommended procedure for early-stage vulvar cancer, but the radioactive technique used as the standard of care has several drawbacks. Our goal was to determine whether an alternative, new method using superparamagnetic iron oxide (SPIO) as the sole tracer is effective and non-inferior to the standard technique. In our study, we showed a 92.3% bilateral detection rate for SLN mapping using SPIO, whereas the detection rate in the groin was 94.3%. SLN mapping failure and complication rates were similar in both study groups. Based on our results, we concluded that SPIO alone is non-inferior to the combination of SPIO and radioactive colloid for SLN detection in patients with early-stage vulvar cancer. In the future, a multicenter, prospective, and comparative study with a significantly more robust study size could address many questions regarding the feasibility and safety of contemporary SLN detection techniques, including radioisotopes, indocyanine green, and superparamagnetic tracers in patients with early-stage disease.

## 1. Introduction

According to the Surveillance, Epidemiology, and End Results (SEER) Program, vulvar cancer (VC) incidence is 2.6 per 100,000 women per year in the United States; however, age-adjusted rates for new cases have been rising by on average 0.7% each year over 2013–2022 [1]. In general, almost two-thirds of cases are diagnosed in the earlier stages (I and II) compared to later stages [2]. Radical surgery, including vulvectomy and inguinofemoral lymphadenectomy (LND), has been recommended to early-stage patients for many years; however, this approach can result in numerous disadvantages such as wound breakdown, infection, and lymphedema, which significantly reduce the quality of life of these mostly older patients. Constantly, groin node status at the time of diagnosis remains the most important indicator of overall survival; therefore, nodal involvement decreases the 5-year overall survival (OS) rate from 90% to 50% [3].

A sentinel lymph node (SLN) detection technique has been developed and standardized in patients with early-stage VC to reduce groin LND complications and to ensure the safety of such procedure with an excellent OS [4,5,6]. The current international guidelines recommend the combination of a radioactive tracer, technetium-99 (Tc-99), and a blue dye for SLN mapping in VC as a standard of care [7,8]. However, this gold standard, with the use of a radioisotope, may be a source of some troublesome issues, such as logistic problems during cooperation with a nuclear medicine unit (especially with those located outside of the managing hospital), some anxiety referring to the ionizing radiation exposure, and the relatively high cost of such a procedure [9]. As a result of these difficulties, alternative SLN mapping techniques have been demonstrated with encouraging outcomes using other tracers such as indocyanine green (ICG) and SPIO [10,11,12,13]. In our previous study we found that a superparamagnetic technique presented a comparable SLN detection rate to that with the use of a radiocolloid tracer; however, the small number of patients included required further confirmatory studies [12].

In the present study, we hypothesized that the exclusive use of SPIO as a tracer for SLN detection is feasible, safe, and non-inferior to the combination of radioisotope (Tc-99) and superparamagnetic tracer use in patients with stage IB vulvar cancer and tumor size ≤ 4 cm. Therefore, we continued the previous SARVU study [12], modified with the SPIO detection technique as a single method for SLN mapping in stage IB patients with VC, especially since during the COVID-19 pandemic our institution has encountered logistical problems with external nuclear medicine suppliers, but had experience in using the SPIO technique alone to identify SLNs in breast cancer patients.

## 2. Materials and Methods

This prospective and observational study concerned carefully selected patients with early-stage VC that were managed in the regional cancer center of southwestern Poland (Department of Oncological Gynecology, Lower Silesian Oncology, Pulmonology and Hematology, in Wroclaw) in the period of 2017 to 2024. The project was approved by the institutional bioethics committee (Wroclaw Medical University Bioethics Committee, registered on 28 November 2017, registration No: 273/2017).

### 2.1. Study Protocol

Detailed information on the superparamagnetic technique, ethical approvals, patient selection, and associated clinical data is provided in our previous publication [12]. Briefly, the study population met the following criteria: diagnosis of primary unifocal squamous VC, apparent in an early stage (namely, stage IB according to the Federation of Gynecology and Obstetrics (FIGO) classification, 2021 update [14]); tumor gross size up to 4 cm in the largest dimension, with preoperative assessment based on magnetic resonance imaging (MRI) examination, including negative groin node status; no evidence of distant metastases on abdominal and chest computer tomography; treatment plan of surgical management including SLN biopsy as part of this plan; and written informed consent of an adult subject for the participation in this study, and available for observation after treatment. Central tumors were diagnosed if they were located within 1 cm of the vulvar midline. Exclusion criteria were the following: any suspicion of metastases, including dubious groin lymph nodes; past inguinal surgery and/or radiotherapy to the affected groin; a metal implant close to the groin; any known intolerance or allergy to iron oxide; an iron excess disease; lack of written consent to participate in this study; mental deterioration; pregnancy; and lactation.

During the initial three years of the project (2017–2019), we included patients in the first study group who underwent combined SLN identification methods (Tc-99 and SPIO). In this part of this study, called the SARVU study, we aimed to test whether the SPIO technique could be effectively used for SLN mapping compared to the standard radioactive detection method [12]. To prove the efficacy of SPIO in this pilot study, we followed a second group of patients treated solely with the superparamagnetic technique for another five years (2020–2024) to demonstrate whether this stand-alone SLN mapping technique is effective and safe compared to the combined procedure, including radiocolloid from the previous period of the study. The design of our study in its second part was based on the assumption that the SPIO mapping method alone was sufficiently effective (which is standard procedure for breast cancer patients at our institution), as well as the logistical issues associated with ensuring the effectiveness of a radioactive colloid tracer during the COVID-19 pandemic, which was problematic at the time (our institution does not have its own nuclear medicine department). The entire surgical procedure was performed in both study groups according to exactly the same template. We did not observe any differences (such as total operation time or number of detected SLNs) between the study groups that could be influenced by the increasing experience with this procedure. For centrally located tumors (in 45 cases included in this study), both groins were mapped for SLN detection, whereas for lateral tumors (in 15 cases included in this study), only 1 groin was evaluated. In the case of SLN mapping failure, inguinofemoral LND was performed in that groin.

The primary endpoint of this investigation was the ratio of patients with successful SLN detection (detection rate). For centrally located tumors, the bilateral SLN detection rate was calculated. Secondary endpoints included the number of SLNs detected per patient, proportion of groins identified with SLN from all evaluated groins (groin detection rate), and proportion of SLNs detected from all removed lymph nodes (nodal detection rate). The malignancy rate (per patient) was calculated as the percentage of patients with metastatic SLNs compared to all patients undergoing SLN mapping. The malignancy rate (per groin) was estimated as the percentage of groins with metastatic SLNs compared to all groins evaluated. All of the above detection and malignancy rates were assessed in the entire study population and in the subgroups studied when a combination of two methods was used for SLN mapping or when a superparamagnetic tracer was used alone. The length (in days) of the total hospitalization time (from admission to discharge) and the postoperative hospitalization stay (from the first day after the surgical procedure to discharge) in the studied cohorts were analyzed. Safety assessments were reported adverse and serious events related to the sentinel lymph node detection procedure in all study patients. The Common Terminology Criteria for Adverse Events (CTCAE) scoring system was used to assess the lymph leakage after SLN biopsy, and Clavien–Dindo classification was used for early (30-day) postoperative complication evaluation in the groin after the SLN surgery [15,16]. The rate of isolated groin recurrences and the probability of disease-free survival (DFS; in months) were analyzed in all study patients who were followed for at least 36 months at our clinic.

Non-inferiority of the stand-alone superparamagnetic technique of SLN mapping in comparison with the combination of radioactive and superparamagnetic tracers was defined as at least an equal SLN detection rate per patient and per groin, as well as lower incidence of postoperative complications in the groin.

### 2.2. Statistical Analyses

This study’s qualitative variables were demonstrated as the absolute value and percentage, while the quantitative data were presented as the mean ± SD (when the data was normal) or median with the range (otherwise). Normally distributed quantitative data were compared between the study groups using Student’s *t*-test, whereas abnormally distributed data were confronted with U Mann–Whitney test. The variance homogeneity (in the *t*-test) was checked using Levene’s test. The relations between categorical variables were analyzed using Fisher’s exact test. Detection rates of both studied SLN mapping methods were expressed as a percentage at the patient level, at the groin level, and at the node level. Similarly, malignancy rate per patient, per groin, and per node was calculated. Survival analysis, concerning the patients with at least 36-month follow-up, was performed using Kaplan–Meyer curves. The survival between the studied groups was compared using the long rank test. In all the analyses, two-tailed testing was used. *p* < 0.05 was considered to indicate a statistically significant difference. The statistical analysis was performed using STATISTICA version 13.3 (TIBCO Software Inc., Palo Alto, CA, USA), and PAST 5 (for Fischer’s exact test) [17].

## 3. Results

A total of 60 patients (representing 110 evaluated groins) were analyzed, of whom 20 underwent the combined SLN detection method (Tc-99 and SPIO) and 40 women were subjected to the superparamagnetic technique alone. The general characteristics of the whole study population are presented in Table 1.

Patients undergoing the combined technique were significantly older (mean age 73.8 years vs. 68.1 years; *p* = 0.046) and had a significantly longer hospitalization stay compared to the SPIO group alone (total hospitalization time 8.6 days; range 3–19 vs. 5.3 days; range 2–10, *p* = 0.005) and postoperative hospitalization time (6.0 days; range 1–17 vs. 2.2 days; range 0–6, *p* = 0.0007). Both study groups did not differ significantly in terms of clinicopathological characteristics, such as body mass index, tumor size (MRI and pathological assessment), stromal infiltration, malignancy grade, and the presence of lymphovascular space involvement (LVSI) (Table 2).

We analyzed groin complications in both study groups, excluding cases with inguinofemoral LND after failed SLN procedure. There were no statistical differences in the incidence of postoperative complications between the two study groups (10% in the combined technique and superparamagnetic alone procedure, *p* > 0.9999), including two cases of CTCAE grade 2 lymph leakage in both groups and one case of Clavien–Dindo Grade IIIa (enlarged lymphocele) in the Tc-99 and SPIO group, and three such cases in the superparamagnetic group alone. Detailed complication descriptions are presented in Table 3.

In both groups, the SLN detection rate was 100%, whereas the bilateral detection rate for centrally located tumors was 84.2% in the combined technique group and 92.3% in the superparamagnetic method alone group. The mean number of SLNs detected per patient was 3.2 (range 1–7) in the entire study group, 3.4 (range 1–6) in the first study group, and 3.1 (range 1–7) in the second group, meaning that the number of SLNs did not differ between the study groups. Additionally, inguinofemoral LND was performed in three cases in the combined group, after successful SLN surgery (due to a surgeon’s discretion) and in four cases after failed one-side SLN mapping. This procedure was carried out in four cases from the superparamagnetic group when SLN detection failed on one side (for a centrally located tumor). Of the 110 evaluated groins of the entire study population, 8 groins failed SLN mapping (7.2%). Among the 40 groins mapped with Tc-99 and SPIO tracers, 4 patients failed SLN detection (10%) and, in the 70 groins mapped with superparamagnetic nanoparticles alone, SLN detection also failed in 4 patients (5.7%). There were nine patients with nodal metastases (SLNs) in the combined group and five patients in the superparamagnetic group. Eighteen metastatic lymph nodes were detected in the combined technique group, while there were nine positive nodes in the superparamagnetic method alone group. The nodal detection rate was calculated to be 42.5% for the Tc-99 and SPIO group and 61.5% for the SPIO-alone group. The SLN malignancy rate per patient was 23.3% (overall) and 45% (Tc-99 and SPIO) and 12.5% (SPIO alone) in the study groups. Of the 110 evaluated groins, 19 were positive (17.2%), with 3 groin metastases detected with LND due to failed SLN mapping (2.7%)—one of these groins was from the combined group (1/40, 2.5%) and two were from the superparamagnetic alone group (2/70, 2.8%). The 3-year isolated groin recurrence rate was 21% for positive groins (4/19) and 2.1% for negative groins (2/91). The above data are presented in Table 4.

During the 36-month follow-up period, we diagnosed groin recurrence in 6 cases of a total number of 35 patients (17.1%) who were followed: 4/19 (21%) in the Tc-99 and SPIO group and 2/16 (12.5%) in the SPIO-alone group. The 3-year DFS was 28.9 months (range 8–36) in the combined group compared to 32.8 months (range 8–36) in the superparamagnetic group. The Kaplan–Meyer curves showed an increased probability of survival for the superparamagnetic alone group (87.5%) compared to the combined group (63.1%), as presented in Figure 1. However, the difference was statistically insignificant (*p* = 0.11) in such a small control population.

## 4. Discussion

In this study, we demonstrated a high detection rate and low failure rate in SLN mapping using SPIO as a single tracer in early-stage VC compared with a combined radioactive and superparamagnetic technique. At least one SLN was detected in all patients studied, whereas in central tumors, the detection rate of bilateral SLN mapping and the groin failure rate were comparable in both study groups. SPIO, as a stand-alone technique for SLN identification in patients with early-stage VC, showed non-inferiority to the combined detection method, including the standard radioactive tracer with respect to groin postoperative complications and groin recurrence rates during a three-year follow-up period.

Superparamagnetic tracing proposes a promising single technique for SLN mapping that has been already implemented in breast cancer patients [18,19,20]. Superparamagnetic solution is a non-radioactive and long-lasting tracer that can be detected with a handheld magnetic probe, eliminating the need for radiocolloid isotopes and reducing logistic challenges while keeping the oncologic safety [18,21]. Therefore, it has the potential to reliably replace the gold standard for SLN mapping in early breast cancer [21]. These observations encouraged us to investigate the feasibility and safety of the superparamagnetic tracing in VC, which is typically a superficial tumor with well-defined lymphatic drainage to the groin. Our findings are consistent with the study by De Valle et al., who showed that the use of SPIO was non-inferior to map SLNs in patients with VC when compared with the radioactive tracer [13]. In their study, the SLN detection rate per patient was 100% for both tracers, while the bilateral detection rate was 100% (SPIO) and 88.9% (Tc-99) [13]. Furthermore, the detection rate in the groin was 100% with the superparamagnetic method and 96.3% with the radioisotope technique compared with our data of 94.3% (superparamagnetic method) and 90% (combined method), respectively. A recent meta-analysis of different SLN detection techniques in patients with VC showed the second highest per-patient detection rate of 95% for a superparamagnetic tracer (after the combination of ICG and Tc-99 mapping, which had a detection rate of 96%), demonstrating its remarkable potential compared with radioactive isotopes, but requires further validation in larger, diverse cohorts [22]. The rapidly increasing use of infrared techniques has contributed to the increase in ICG application for SLN detection in VC due to its real-time visualization of lymphatic mapping, resulting in excellent SLN detection and precise dissection [10,23]. Di Donna et al. reported a detection rate for ICG of 91.9% per patient and 94,8% per groin in their meta-analysis [24]. Another large meta-analysis demonstrated the pooled detection rate of SLN surgery using ICG as 87% (per patient) and 88% (per groin) [22]. In our study, the superparamagnetic technique demonstrated better detection rates than an ICG alone (100% per patient, 94.3% per groin). Therefore, this option seems to be the only method to be used alone, as ICG as a sole tracer for SLN detection in VC should be considered only in combination with Tc-99 nanocolloids [25]. On the other hand, the SLN detection rate with ICG alone has been found to be highly effective at 96.3%, with a failure rate of SLN detection in the groin of 3.7% compared to 5.7% in our study [11]. In their large retrospective analysis, these authors showed that 14,6% of patients with VC that underwent the SLN tracing with the combination of Tc-99 and ICG failed to map with the radioisotope but were mapped with ICG alone. These discrepant data suggest that the new techniques of SLN mapping with the use of ICG and SPIO in early VC patients require large prospective, multi-center, and comparative studies to establish the most effective and safest contemporary standard of care.

SLN mapping in vulvar cancer is considered a safe procedure, with few short-term postoperative problems, such as lymphoma formation and wound infection or breakdown. The complication rate in our study, in both studied groups, was 10%. Robison et al. reported a complication rate of 17.5% in the inguinal wound after SLN surgery (cellulitis, abscess, lymphoceles, lymphedema, and leg pain) [26]. An increased number of postoperative complications (25%) was observed in the combination of standard tracers (Tc-99 and blue dye) compared with and ICG tracing (4.2%). No long-term complications occurred [27]. Similarly, we observed no long-term complications in our study cohort. We observed a significantly shorter hospital stay in the superparamagnetic group compared with the combined group, but the latter study group was significantly older and required more intensive care. The median postoperative hospital stay in the SPIO group (2 days) was similar to that in another study [28]. However, the total hospitalization time of our study population was significantly longer than in other studies [28,29]. We explained these differences by taking into account the logistical issues associated with the use of Tc-99.

During a 3-year observation, we noted a groin recurrence rate of 2.1% in our patients with SLN negative status. Exactly the same isolated groin recurrence rate in early-stage vulvar cancer with negative SLNs was reported by a Danish nation-wide study [30]. In a large, prospective study (GROINNS-V-I), eight groin recurrences (2.9%) were reported, with a median follow-up of 35 months [4]. Similarly, Robison et al. observed a 4.7% groin recurrence rate in long-term follow-up (58 months) after SLN biopsy alone [26]. The Memorial Sloane Kettering Cancer Center group demonstrated in their multi-year retrospective analysis an even lower rate of isolated groin recurrence in vulvar cancer patients with negative sentinel lymph node biopsy (rate 1.2%; 2 relapses among 169 evaluated groins), despite having a follow-up period of only 24 months [11]. In terms of survival, we observed better outcomes in the superparamagnetic group, with a median 3-year DFS of 32.8 months and probability of survival of 87.5% after 36 months of follow-up, compared with the combined tracer group, but these data were statistically insignificant. Likewise, Hermann et al. found no differences in 3-year overall survival between vulvar cancer patients undergoing SLN biopsy alone, SLN biopsy, and inguinofemoral LND or LND, with survival rates of 86.3%, 77.9%, and 82.1%, respectively [29]. In a large, nation-wide retrospective analysis, 3-year overall survival and disease-specific survival for SLN-negative patients was 84% and 93%, respectively, which is consistent with our results. However, it should be remembered that our survival analysis focused on isolated groin recurrence, whereas the Danish study contained all recurrence sites, including the vulva [30].

We presented cases of patients treated at a single institution who had experience with SLN mapping in breast cancer patients with the use of superparamagnetic nanoparticles. The study population was carefully selected to ensure maximum homogeneity and comparability. We provided a consistent surgical template for the procedure to warrant reproducibility and consistency. VC recurrence typically occurs within the first 2 years of follow-up [31]; therefore, our analysis of isolated groin recurrence and survival probability after SLN mapping biopsy covered 36 months. In the context of the feasibility and safety of SPIO detection of SLNs in groins, we analyzed for the first time, to our knowledge, the incidence of complications and hospitalization time in patients with VC treated with superparamagnetic tracking, compared with the combined use of radiocolloid and SPIO. The SPIO technique is relatively easy to use compared to radiotracers, and the learning curve is short, especially in the institutions that already have a handhold probe magnetometer and have experience in its use in SLN biopsy for other malignancies.

Our study also has limitations. The study population was limited, particularly in the combined group, which could have biased our statistical calculations, indicating limited power. Fortunately, the study groups were comparably balanced with respect to clinicopathological risk factors, which should not affect detection rates. Furthermore, analysis of the relationships of categorical variables, taking into account these identified risk factors, revealed no significant differences between the study groups, except for age (a significantly older population in the combined group). Finally, we did not perform inguinofemoral LND after successful SLN detection. Therefore, we could not calculate the sensitivity of the superparamagnetic technique or a negative predictive value, but relied on the groin recurrence rate as a surrogate for the accuracy of SLN detection with a superparamagnetic tracer, without exposing the study population to the morbidity associated with complete inguinal LND unless necessary due to failed SLN biopsy. Nevertheless, a multi-center prospective study of SPIO as an SLN mapping technique followed by inguinofemoral LND should be performed in the future to verify its sensitivity and negative predictive value.

## 5. Conclusions

The use of superparamagnetic tracing alone in the SLN detection of early-stage VC patients is not inferior to the present standard of care. In this preliminary study, it appears to be an excellent alternative to a radioactive tracer SLN mapping, with comparable SLN detection rates and similar rates of groin disease recurrence and survival probability, with no difference in the occurrence of postoperative complications in the groin. Institutions without a nuclear medicine unit but with experience in using the superparamagnetic method in breast cancer for SLN mapping may benefit from this innovative technique in patients with VC. Our findings support the need for further research on the exclusive use of the superparamagnetic technique for SLN mapping, which could eliminate the drawbacks of the current standard of radioisotope use; however, it is too early to recommend this method as an alternative due to the lack of validation studies. The SPIO technique has potential, especially in countries where there are no nuclear medicine centers; therefore, the ethnic background of subjects undergoing this new method of SLN detection may be of great interest. We believe it is time to reassess whether the recommended standard of radioactive tracer tracking for SLN detection in patients with VC is still the gold standard. A multi-center, prospective, and comparative study with a significantly larger sample size could answer many questions regarding the feasibility and safety of current SLN detection techniques, including Tc-99, ICG, and SPIO tracers.

## Figures and Tables

**Figure 1 cancers-17-03722-f001:**
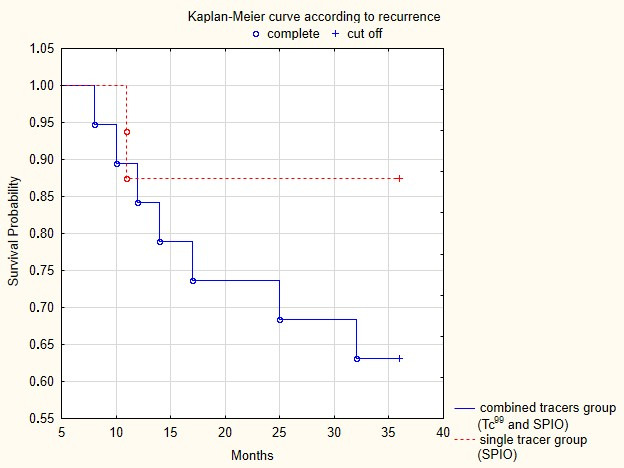
Disease-free survival in the two study groups over a 36-month follow-up period. Kaplan–Meyer curves for vulvar cancer recurrence (*p* = 0.11).

**Table 1 cancers-17-03722-t001:** General characteristics of the study population.

Clinical Feature	All Study Patients (*n* = 60)	Combined Group (Tc-99 and SPIO) (*n* = 20)	SPIO-alone Group (*n* = 40)
Average age (years)	70.0	73.8	68.1
Average BMI (kg/m^2^)	29.8	30.8	29.3
ECOG performance status 1	32 (53.3%)	8 (40%)	24 (60%)
ECOG performance status 2	28 (46.7%)	12 (60%)	16 (40%)
Average tumor size (mm)	23.1	26.6	21.3
Tumor grade 1	28(46.7%)	7 (35%)	21 (52.5%)
Tumor grade 2	31 (51.6%)	13 (65%)	18 (45%)
Tumor grade 3	1 (1.7%)	0	1 (2.5%)
Tumor location central	45 (75%)	19 (95%)	26 (65%)
Tumor location lateral	15 (25%)	1 (5%)	14 (35%)
Average stromal infiltration (mm)	4.3	4.5	4.1
LVSI positive	5 (8.3%)	2 (10%)	3 (7.5%)
LVSI negative	55 (91.7%)	18 (90%)	37 (92.5%)
Unilateral sentinel lymph node assessment	18 (30%)	4 (25%)	14 (35%)
Bilateral sentinel lymph node assessment	42 (70%)	16 (75%)	26 (65%)

Abbreviations: Tc-99—technetium-99; SPIO—superparamagnetic iron oxide; BMI—body mass index; MRI—magnetic resonance imaging; LVSI—lymphovascular space involvement.

**Table 2 cancers-17-03722-t002:** Comparison of clinical and pathological features of the studied groups.

Studied Feature	Combined Group (Tc-99 and SPIO)	SPIO-Alone Group	*p* Value
Age (years) (mean ± SD)	73.8 (8.9)	68.1 (10.7)	0.0468 ^t^
BMI (kg/m^2^) (median (range))	30.8 (22–45)	29.3 (18–36)	0.1185 ^m^
Total hospitalization time (days) (median (range))	8.6 (3–19)	5.3 (2–10)	0.0057 ^m^
Postoperative hospitalization time (days) (median (range))	6.0 (1–17)	2.2 (0–6)	0.0007 ^m^
MRI maximum tumor size (mm) (median (range))	26.6 (0–40)	21.3 (0–40)	0.1548 ^m^
Pathological maximum tumor size (mm) MRI maximum tumor size (mm) (median (range))	27.6 (3–50)	22.5 (5–50)	0.1069 ^m^
Stromal infiltration (mm) (median (range))	4.5 (1–16)	4.1 (1–16)	0.1245 ^m^
*G1* tumors (*n*)	7	21	0.3001 ^f^
*G2* tumors (*n*)	13	18	
*G3* tumors (*n*)	0	1	
LVSI positive (*n*)	2	18	>0.9999 ^f^
LVSI negative (*n*)	3	37	

Abbreviations: Tc-99—technetium-99; SPIO—superparamagnetic iron oxide; BMI—body mass index; ECOG—eastern cooperative oncology group; G—grading; LVSI—lymphovascular space involvement; ^t^—t-Student test; ^m^—Mann–Whitney test; ^f^—Fisher’s exact test. The red color in the table is used to emphasize the statistical significance of the result.

**Table 3 cancers-17-03722-t003:** Comparison of groin complications after SLN detection procedure in the studied groups.

Complications After Sentinel Lymph Node Procedure	Combined Group (Tc-99 and SPIO)	SPIO-Alone Group	How Was It Managed?
Lymph leakage—CTCEA Grade 1 (*n*)	0	2	observation and conservative treatment
Lymph leakage—CTCEA Grade 2 (*n*)	2	2	drainage and inguinal wound lavage
Lymphocele—Clavien–Dindo Grade I (*n*)	1	1	observation and conservative treatment
Lymphocele—Clavien–Dindo Grade IIIa (*n*)	1	3	surgical decompression of lymphocoele with catheter

Abbreviations: Tc-99—technetium-99; SPIO—superparamagnetic iron oxide; CTCEA—Common Terminology Criteria for Adverse Events.

**Table 4 cancers-17-03722-t004:** SLN detection parameters of the whole study population and study subgroups depending on the method used.

Detection Parameters (%)	All Studied Patients	Combined Group (Tc-99 and SPIO)	SPIO-Alone Group
SLN detection rate (overall)	100	100	100
SLN bilateral detection rate (refers to centrally located tumors)	88.2	84.2	92.3
Groin detection rate	92.8	90	94.3
Nodal detection rate	53.1	42.5	61.5
SLN malignancy rate (per patient)	23.3	45	12.5
SLN malignancy rate (per groin)	17.2	30	10

Abbreviations: SLN—sentinel lymph node; Tc-99—technetium-99; SPIO—superparamagnetic iron oxide.

## Data Availability

The data will be made available on request.

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
