# Peer review of "Superparamagnetic Iron Oxide Used Alone Is Non-Inferior to the Combination of Radioactive Tracer and Superparamagnetic Iron Oxide in Detecting Sentinel Lymph Nodes in Early-Stage Vulvar Cancer"

_cancers, 2025, doi:10.3390/cancers17233722_

Round 1
Reviewer 1 Report
Comments and Suggestions for Authors
This article compares the differences between SPIO alone and in combination in sentinel lymph node detection in vulvar cancer. The analytical methods are appropriate and the structure is reasonable.
However, several questions require detailed explanation.
- Existing articles have evaluated the efficacy of SPIO alone, but the authors chose a combination of SPIO and Tc-99. Please explain the reasons for choosing this combination. This directly relates to the significance of this study.
- The Kaplan-Meier curve analysis results are misleading. Figure 1 shows that the patient group using SPIO alone had a better prognosis, although it did not reach statistical significance.
a) Is the small number of cases representative of the phenomenon?
b) lack of clinical information about the group, Is he output related to the method of SPIO use?
Author Response
Reviewer #1
Open Review
(x) I would not like to sign my review report
( ) I would like to sign my review report
Quality of English Language
( ) The English could be improved to more clearly express the research.
(x) The English is fine and does not require any improvement.
|
Yes |
Can be improved |
Must be improved |
Not applicable |
|
|
Does the introduction provide sufficient background and include all relevant references? |
( ) |
( ) |
(x) |
( ) |
|
Is the research design appropriate? |
(x) |
( ) |
( ) |
( ) |
|
Are the methods adequately described? |
( ) |
(x) |
( ) |
( ) |
|
Are the results clearly presented? |
(x) |
( ) |
( ) |
( ) |
|
Are the conclusions supported by the results? |
(x) |
( ) |
( ) |
( ) |
|
Are all figures and tables clear and well-presented? |
(x) |
( ) |
( ) |
( ) |
Comments and Suggestions for Authors
This article compares the differences between SPIO alone and in combination in sentinel lymph node detection in vulvar cancer. The analytical methods are appropriate and the structure is reasonable.
However, several questions require detailed explanation.
- Existing articles have evaluated the efficacy of SPIO alone, but the authors chose a combination of SPIO and Tc-99. Please explain the reasons for choosing this combination. This directly relates to the significance of this study.
- The Kaplan-Meier curve analysis results are misleading. Figure 1 shows that the patient group using SPIO alone had a better prognosis, although it did not reach statistical significance.
- a) Is the small number of cases representative of the phenomenon?
- b) lack of clinical information about the group, Is he output related to the method of SPIO use?
Response to Reviewer #1
Thank you very much for your work and comments that helped us to improve this manuscript a lot. All Reviewers’ comments and suggestions were analysed and taken into account. All corrections made in the manuscript have been highlighted in yellow and additionally, crossed out if deleted. Figure 1 has been corrected and added to this version of the manuscript. The English language of the manuscript has been reviewed and improved by a native speaker.
The answers to the Reviewers’ comments can be found below as a point-by-point response to each of the Reviewers’ comment.
- Existing articles have evaluated the efficacy of SPIO alone, but the authors chose a combination of SPIO and Tc-99. Please explain the reasons for choosing this combination. This directly relates to the significance of this study.
Thank you for pointing this out. At first, we studied the accuracy of a novel method for SLN detection in early-stage vulvar cancer patients using the SPIO technique compared to the standard of care (radiocolloid technique wit Tc99) introducing both techniques at the same time (combination group). We published the preliminary results of our study (SARVU) – ref. 12. Then, we decided to continue this research to enlarge the study population however, it happened over the years of COVID-19 pandemia and we encountered huge logistical problems with external nuclear medicine collaborators (our institution does not have such facility). On the other hand, surgical oncologists in Breast Cancer Clinic of our hospital have great experience in using SPIO technique alone in SLN mapping. That motivated us to continue the previous study but without combination of tracers (Tc99 and SPIO), just using a single tracer for SLN mapping (SPIO alone group) and compare the results of those two studied groups. We included the proper explanations concerning the selection of the study groups in the manuscript (Introduction; lines 88-92; Material and Methods; lines 118-125)
- The Kaplan-Meier curve analysis results are misleading. Figure 1 shows that the patient group using SPIO alone had a better prognosis, although it did not reach statistical significance.
- a) Is the small number of cases representative of the phenomenon?
- b) lack of clinical information about the group, Is he output related to the method of SPIO use?
Yes, we agree with your comment that in Figure 1 K-M curves are diverging what could suggest that the SPIO group has a significant better prognosis. We talked this figure with our statistician who stated that this was just a tendency – favorable for the SPIO group – however, insignificant. This is the result of small number of representative cases from both group that were included to this analysis comprising cases with at least 36 months of follow-up. We corrected Fig. 1 giving there proper clinical information about the study groups. We explained this tendency of better survival favorable for the SPIO alone group in the K-M curves but without significance due to the small control group. (Results; lines 235-238).
Reviewer 2 Report
Comments and Suggestions for Authors
The presented manuscript is of interest. However there are some points that require clarification and revision
1 In the Abstract section (Objective) the phrase “to evaluate.. and non-inferiority” gives the reader the sense that the authors have pre-decided the outcome, suggesting that there is bias. I suggest a careful rephrasing in that point.
2 There are some typos throughout the manuscript (eg. double space at line 64 that should be fixed.
3 The authors mention in the final paragraph of the introduction “we hypothesized that..”. Can you justify why you hypothesized this? It remains unclear for the reader. The rationale should be explained.
4 A large part of the discussion (in particular after line 255) is just a row of numbers that belongs in the results, not in the discussion section.
5 the learning curve of SPIO should also be discussed
6 Overall the discussion should be a little more dense and not mentioning the results anew serving as a second result section
7 In the conclusion section, the statement “our findings support the exclusive use of.. radioisotope use” seems bold. This could be said after a multi-center, prospective study has been conducted (as mentioned a few lines later” and not after a study with a rather limited sample number. I suggest a more careful wording so as not to be misleading for the readers. In parallel it would be interesting to know the ethnic background of the subjects since it could be valuable for future studies in terms of external validity and generalizability.
Author Response
Reviewer #2
Open Review
(x) I would not like to sign my review report
( ) I would like to sign my review report
Quality of English Language
(x) The English could be improved to more clearly express the research.
( ) The English is fine and does not require any improvement.
|
Yes |
Can be improved |
Must be improved |
Not applicable |
|
|
Does the introduction provide sufficient background and include all relevant references? |
( ) |
(x) |
( ) |
( ) |
|
Is the research design appropriate? |
(x) |
( ) |
( ) |
( ) |
|
Are the methods adequately described? |
(x) |
( ) |
( ) |
( ) |
|
Are the results clearly presented? |
( ) |
(x) |
( ) |
( ) |
|
Are the conclusions supported by the results? |
( ) |
( ) |
(x) |
( ) |
|
Are all figures and tables clear and well-presented? |
(x) |
( ) |
( ) |
( ) |
Comments and Suggestions for Authors
The presented manuscript is of interest. However there are some points that require clarification and revision
Response to Reviewer #2
Thank you very much for your work and comments that helped us to improve this manuscript a lot. All Reviewers’ comments and suggestions were analysed and taken into account. All corrections made in the manuscript have been highlighted in yellow and additionally, crossed out if deleted. Figure 1 has been corrected and added to this version of the manuscript. The English language of the manuscript has been reviewed and improved by a native speaker.
The answers to the Reviewers’ comments can be found below as a point-by-point response to each of the Reviewers’ comment.
1 In the Abstract section (Objective) the phrase “to evaluate.. and non-inferiority” gives the reader the sense that the authors have pre-decided the outcome, suggesting that there is bias. I suggest a careful rephrasing in that point.
Thank you for pointing this out. We rephrased this sentence according to your comment (Abstract; line 36).
2 There are some typos throughout the manuscript (eg. double space at line 64 that should be fixed.
Thank you for pointing this out. We carefully looked through the manuscript and corrected those typos.
3 The authors mention in the final paragraph of the introduction “we hypothesized that..”. Can you justify why you hypothesized this? It remains unclear for the reader. The rationale should be explained.
We agree with this comment and included a proper explanation to this paragraph accordingly (Introduction; lines 88-92).
4 A large part of the discussion (in particular after line 255) is just a row of numbers that belongs in the results, not in the discussion section.
Thank you very much for pointing this out. Indeed, this part of discussion should not be a repetition of the Result section, therefore we have changed it completely (Discussion; lines 246-259 and 270).
5 the learning curve of SPIO should also be discussed
We agree with your comment. The proper statement has been included in the Discussion part of the manuscript (lines 340-343).
6 Overall the discussion should be a little more dense and not mentioning the results anew serving as a second result section
We agree therefore we rephrased the discussion part removing repetition data from the Results section and making this section more consistent (Discussion; lines 246-259).
7 In the conclusion section, the statement “our findings support the exclusive use of.. radioisotope use” seems bold. This could be said after a multi-center, prospective study has been conducted (as mentioned a few lines later” and not after a study with a rather limited sample number. I suggest a more careful wording so as not to be misleading for the readers. In parallel it would be interesting to know the ethnic background of the subjects since it could be valuable for future studies in terms of external validity and generalizability.
Thank you for pointing out this important conclusion. We completely agree with your remarks as our study was just a preliminary one and definitely, SPIO technique for SLN mapping in vulvar cancer patients requires prospective, multi-center studies for validation. Therefore, we have rephrased the Conclusion section and included the reviewers’ comments (Conclusions; lines 360 and 366-373).
Reviewer 3 Report
Comments and Suggestions for Authors
This prospective study investigated whether superparamagnetic iron oxide (SPIO) nanoparticles alone can effectively detect sentinel lymph nodes (SLNs) in early-stage vulvar cancer compared with the standard combined method using technetium-99 (Tc-99) and SPIO. SLN detection was successful in all patients, with similar bilateral detection rates and complication profiles between groups. The SPIO-only group showed shorter hospitalization time and comparable 3-year disease-free survival and recurrence rates. Overall, the findings support SPIO as a safe, feasible, and non-inferior alternative to radioactive tracers for SLN mapping in vulvar cancer. This is an interesting topic in Oncology. However, several sections of the manuscript require revision and clarification before it can be considered for publication.
- The study includes 60 patients, with only 20 in the combined-tracer group. Please clarify whether a formal sample-size was performed to support statistical rigor.
- Since the combined-tracer group was enrolled earlier and the SPIO-only group later, temporal or practice-related changes in surgical experience may influence outcomes. Please discuss how this was controlled.
- Without complete inguinofemoral lymphadenectomy in SLN-negative cases, the accuracy of SPIO cannot be directly validated. Please elaborate on how this limitation affects interpretation and future directions.
- The SPIO group had a significantly shorter postoperative stay. Were there institutional workflow differences over time that may explain this outcome beyond tracer choice?
- Bilateral detection and recurrence analysis could benefit from survival modeling including clinical covariates. If not feasible, please justify the current analytical approach.
- Complication rates appear similar, but the SPIO-only group showed more Grade IIIa lymphoceles (3 vs. 1). Please clarify whether this difference may be clinically meaningful despite the small sample.
- The similarity index of the manuscript is currently 30%. It is recommended to reduce it to 20% to ensure originality and avoid potential plagiarism concerns.
Author Response
Reviewer # 3
Open Review
(x) I would not like to sign my review report
( ) I would like to sign my review report
Quality of English Language
( ) The English could be improved to more clearly express the research.
(x) The English is fine and does not require any improvement.
|
Yes |
Can be improved |
Must be improved |
Not applicable |
|
|
Does the introduction provide sufficient background and include all relevant references? |
(x) |
( ) |
( ) |
( ) |
|
Is the research design appropriate? |
( ) |
(x) |
( ) |
( ) |
|
Are the methods adequately described? |
( ) |
(x) |
( ) |
( ) |
|
Are the results clearly presented? |
( ) |
(x) |
( ) |
( ) |
|
Are the conclusions supported by the results? |
( ) |
(x) |
( ) |
( ) |
|
Are all figures and tables clear and well-presented? |
( ) |
(x) |
( ) |
( ) |
Comments and Suggestions for Authors
This prospective study investigated whether superparamagnetic iron oxide (SPIO) nanoparticles alone can effectively detect sentinel lymph nodes (SLNs) in early-stage vulvar cancer compared with the standard combined method using technetium-99 (Tc-99) and SPIO. SLN detection was successful in all patients, with similar bilateral detection rates and complication profiles between groups. The SPIO-only group showed shorter hospitalization time and comparable 3-year disease-free survival and recurrence rates. Overall, the findings support SPIO as a safe, feasible, and non-inferior alternative to radioactive tracers for SLN mapping in vulvar cancer. This is an interesting topic in Oncology. However, several sections of the manuscript require revision and clarification before it can be considered for publication.
Response to Reviewer #3
Thank you very much for your work and comments that helped us to improve this manuscript a lot. All Reviewers’ comments and suggestions were analysed and taken into account. All corrections made in the manuscript have been highlighted in yellow and additionally, crossed out if deleted. Figure 1 has been corrected and added to this version of the manuscript. The English language of the manuscript has been reviewed and improved by a native speaker.
The answers to the Reviewers’ comments can be found below as a point-by-point response to each of the Reviewers’ comment.
- The study includes 60 patients, with only 20 in the combined-tracer group. Please clarify whether a formal sample-size was performed to support statistical rigor.
Thank you very much for this comment. This study was designed as a preliminary research before a larger study (preferable multicenter research) with an adequate statistical power. Therefore, a formal sample-size assessment was not performed to support statistical report. Additionally, during most time of the study we encountered problems with some logistic issues over the years of COVID-19 pandemia and we encountered huge logistical problems with external nuclear medicine collaborators (our institution does not have such facility). On the other hand, surgical oncologists in Breast Cancer Clinic of our hospital have great experience in using SPIO technique alone in SLN mapping. That motivated us to continue the previous SARVU study but without of combination (Tc99 and SPIO), just using a single tracer for SLN mapping (SPIO alone group) and compare the results of those two studied groups. We included the proper explanations concerning the selection of the study groups in the manuscript, underlining its weakness (Introduction; lines 88-92; Material and Methods; lines 118-125; Discussion; lines 344-350).
- Since the combined-tracer group was enrolled earlier and the SPIO-only group later, temporal or practice-related changes in surgical experience may influence outcomes. Please discuss how this was controlled.
Thank you very much for this comment. Indeed, the increasing experience of surgeons could influence the results, favoring the latter study group. However, we kept very strictly to the surgical template in both groups and thus did not observe the result differences that might be impacted by this study pattern. We included the proper explanation in the manuscript (Material and Methods; lines 125-129; Discussion; lines 331-335).
- Without complete inguinofemoral lymphadenectomy in SLN-negative cases, the accuracy of SPIO cannot be directly validated. Please elaborate on how this limitation affects interpretation and future directions.
Thank you for pointing this out. We emphasized this fact in the discussion section expressing this weakness of the study and what we assumed to solve this problem (Discussion; lines 350-355). We also stated that only a multi-center, prospective study of SLN followed by LND is mandatory to assess this method sensitivity and negative predictive value (Discussion; lines 355-357).
- The SPIO group had a significantly shorter postoperative stay. Were there institutional workflow differences over time that may explain this outcome beyond tracer choice?
Thank you for pointing this out. We did not change our institutional workflow during the study. We explain this difference with the fact that the combined group was significantly older thus, requiring more intensive care which just took more time. We included the proper statement in the manuscript (Discussion; lines 304-307).
- Bilateral detection and recurrence analysis could benefit from survival modeling including clinical covariates. If not feasible, please justify the current analytical approach.
We agree with this comment however, in our study the small size of the study groups could not allow to analyze bilateral detection rate and recurrence rate in correlation with clinical covariates. Therefore, detection rates of both studied SLN mapping methods were expressed as a percentage at the patient level, at the groin and, at the node level while groin recurrence rate at the patient level. The survival between the studied groups was compared using the long rank test separately. These statistical analyses were described in the Materials and Methods part of the manuscript (lines 158-173).
- Complication rates appear similar, but the SPIO-only group showed more Grade IIIa lymphoceles (3 vs. 1). Please clarify whether this difference may be clinically meaningful despite the small sample.
Thank you very much for this comment. In our opinion, this very small sample of Grade IIIa lymphoceles definitely has no clinical impact as it is too low to have any significance. If we consider Grade IIIa lymphocele of 3 cases in the SPIO group (n=40) and 1 case in the combined group (n=20), it means that its incidence is 7.5% in the SPIO alone group and 5% in the combined group which are actually almost the same. The proper comment is included in the Results part (lines 194-200).
- The similarity index of the manuscript is currently 30%. It is recommended to reduce it to 20% to ensure originality and avoid potential plagiarism concerns.
Thank you for your comment, we agree with it however, some sections in this manuscript have been paraphrased from our previous article (ref. 12) which was related to the present work, but they have all been cited in the reference section. Therefore, we do not consider this high similarity index as plagiarism. Nevertheless, we did our best to lower this index by rephrasing the words and sentences from our previous publications.
Round 2
Reviewer 1 Report
Comments and Suggestions for Authors
The author addressed my questions. However, I don't quite understand the answer to the first question, why choose a combination? This relates to the overall theme of the article. Since using a combination is less effective than using individual items, why choose a combination? The author could more reasonably explain the reasons for designing the clinical study in this way.
Author Response
Reviewers’ comments and responses – ROUND TWO
Thank you very much for your time and valuable comments. All Reviewers’ comments and suggestions were analysed and taken into account. All corrections made in the manuscript have been highlighted in yellow and additionally, crossed out if deleted.
Figure 1 was corrected and added to the revised manuscript during Round 1 review. The English language of the manuscript was reviewed and improved by an English native speaker during Round 1 review.
The answers to the Reviewers’ comments can be found below as a point-by-point response to each of the Reviewers’ comment.
Reviewer # 1:
The author addressed my questions. However, I don't quite understand the answer to the first question, why choose a combination? This relates to the overall theme of the article. Since using a combination is less effective than using individual items, why choose a combination? The author could more reasonably explain the reasons for designing the clinical study in this way.
Thank you very much for pointing this out. Indeed, our explanation for selecting the study groups was akward. First, we compared in a pilot study two tracers administered independently (radioactive colloid – Tc99 and SPIO) demonstratinging their comparable efficacy and accuracy. Those results were published as a preliminary study (named SARVU study): Jedryka, M.A.; Klimczak, P.; Kryszpin, M.; Matkowski, R. Superparamagnetic Iron Oxide: A Novel Tracer for Sentinel Lymph Node Detection in Vulvar Cancer. International Journal of Gynecological Cancer 2020, 30, 1280–1284, doi:10.1136/ijgc-2020-001458.
We decided to continue this study with a larger population and longer observation time to investigate feasibility of a new method (SPIO) however, the design of our study in its second part was based on the assumption that the SPIO mapping method alone was sufficiently effective (which is standard procedure for breast cancer patients at our instititution), as well as the logistical issues associated with ensuring the of a radioactive colloid tracer during COVID-19 pandemic, which was problematic at the time (our institution does not have its own nuclear medicine department). For the above reasons we had two study population to compare: before the pandemic (with two techniques of SLN mapping used at the same time) and during the pandemic (with just one technique used – namely, SPIO).
We did our best to explain the reasons for designing our study in this way, including them in the manuscript (Materials and Methods, lines: 118-134).
Reviewer 2 Report
Comments and Suggestions for Authors
Thank you for answering my comments.
Author Response
Reviewers’ comments and responses – ROUND TWO
Thank you very much for your time and valuable comments. All Reviewers’ comments and suggestions were analysed and taken into account. All corrections made in the manuscript have been highlighted in yellow and additionally, crossed out if deleted.
Figure 1 was corrected and added to the revised manuscript during Round 1 review. The English language of the manuscript was reviewed and improved by an English native speaker during Round 1 review.
The answers to the Reviewers’ comments can be found below as a point-by-point response to each of the Reviewers’ comment.
Reviewer # 2:
Thank you for answering my comments.
Thank you very much for your time and commitment in reviewing this manuscript.
Reviewer 3 Report
Comments and Suggestions for Authors
The author has revised the manuscript in accordance with the reviewers’ suggestions; therefore, the reviewers have accepted it for publication.
Author Response
Reviewers’ comments and responses – ROUND TWO
Thank you very much for your time and valuable comments. All Reviewers’ comments and suggestions were analysed and taken into account. All corrections made in the manuscript have been highlighted in yellow and additionally, crossed out if deleted.
Figure 1 was corrected and added to the revised manuscript during Round 1 review. The English language of the manuscript was reviewed and improved by an English native speaker during Round 1 review.
The answers to the Reviewers’ comments can be found below as a point-by-point response to each of the Reviewers’ comment.
Reviewer # 3:
The author has revised the manuscript in accordance with the reviewers’ suggestions; therefore, the reviewers have accepted it for publication.
Thank you very much for your time and commitment in reviewing this manuscript.